# CryoHype: Transformer-based hypernetworks for extreme heterogeneity in cryo-EM

## Abstract

Cryo-electron microscopy (cryo-EM) is an indispensable technique for determining the 3D structures of dynamic biomolecular complexes. While typically applied to image a single molecular species, cryo-EM holds great potential for structure determination of many targets simultaneously in a high-throughput fashion. However, existing methods typically focus on modeling *conformational heterogeneity* within a single or a few structures and are not designed to resolve *compositional heterogeneity* arising from mixtures of many distinct molecular species. To address this challenge, we propose CryoHype, a transformer-based hypernetwork for cryo-EM reconstruction that dynamically adjusts the weights of an implicit neural representation conditioned on each particle image. CryoHype establishes a new state-of-the-art on the challenging Tomotwin-100 dataset for compositional heterogeneity in CryoBench. We further introduce Sim2Struct-1000, a new synthetic dataset for compositional heterogeneity with 10 times more structures than previous datasets, where CryoHype improves $\text{FSC}_{\text{AUC}}$ by 67%. Together, these advances establish transformer hypernetworks as a scalable approach for extreme heterogeneity in cryo-EM reconstruction.

## 1 Introduction

Single particle cryo-electron microscopy (cryo-EM) has emerged as an essential tool to resolve the 3D structures of macromolecular complexes at atomic resolution (Nakane et al., 2018; Yip et al., 2020). Unlike other structure determination methods or structure prediction methods, cryo-EM can experimentally probe the dynamics of large macromolecular complexes in near-native states.

In cryo-EM imaging, an aqueous solution of biomolecular complexes is flash frozen and imaged using an electron microscope. Each image contains an extremely noisy, low signal-to-noise ratio (SNR) projection of a complex in an unknown orientation, making the inverse problem of 3D reconstruction especially challenging. Traditionally, cryo-EM is typically used to resolve a single or a few structures from a purified sample. Yet the technique can, in principle, be used to capture more complex scenarios, including heterogeneous mixtures, unpurified samples, or cellular lysates to determine multiple structures in a high-throughput fashion (Ho et al., 2020; Rabuck-Gibbons et al., 2022; Jeon et al., 2024). Resolving this discrete heterogeneity due to the presence of multiple structures, termed *compositional heterogeneity*, is a major challenge for cryo-EM reconstruction algorithms. In this work, we focus on expanding the capabilities of cryo-EM imaging by tackling the computational challenge of capturing extreme, large-scale compositional heterogeneity.

Classical reconstruction algorithms handle compositional heterogeneity by using algorithms such as expectation-maximization to sort images into a small, predefined number of discrete classes, typically fewer than 10. However, these approaches struggle to handle large-scale heterogeneity due to computational limitations. Neural-based lines of work use either an autoencoding (Zhong et al., 2020; 2021) or encoder-free autodecoding (Punjani & Fleet, 2021; Levy et al., 2024a;b) approach. Volumes are representing using linear combinations of voxel arrays (Punjani & Fleet, 2021; Kimanius et al., 2022) or neural representations (Zhong et al., 2020; 2021; Levy et al., 2024a;b). While these methods may be sufficient for modeling a small number of structures or compositional heterogeneity found within a single biomolecular complex, excessive parameter sharing between the representations of different structures causes them to inadequately capture extreme compositional heterogeneity, limiting the resolution and diversity of the reconstructed structures.

To overcome these limitations, we introduce CryoHype, an autoencoding transformer-based hypernetwork method that can resolve extreme compositional heterogeneity. Using a hypernetwork (Ha et al., 2016) encoder allows the model to dynamically adapt the weights of the neural representations to different structures, reducing parameter sharing and increasing expressivity compared to past methods. The vision transformer (Dosovitskiy, 2020) (ViT) architecture for the hypernetwork provides scalable and parameter-efficient weight generation. In order to robustly evaluate CryoHype, we introduce `Sim2Struct-1000`, a dataset for extreme compositional heterogeneity with 10 times more structures than previous datasets for compositional heterogeneity (Jeon et al., 2024), and propose new real-space metrics from 3D shape analysis that complement traditional Fourier Shell Correlation (FSC)-based metrics. We demonstrate that our method scales neural-based methods to extreme compositional heterogeneity with state-of-the-art performance on a variety of metrics. We therefore make the following contributions:

- We identify excessive parameter sharing as a limitation of current neural methods, and propose CryoHype, a transformer-based hypernetwork model for heterogeneous cryo-EM reconstruction;

- We demonstrate that our method can reconstruct datasets containing extreme large-scale compositional heterogeneity up to 1000 distinct structures;

- We propose `Sim2Struct-1000`, a new large-scale dataset for compositional heterogeneity that is an order of magnitude larger than previous datasets;

- We propose two new metrics from 3D shape analysis that complement traditional FSC-based metrics to provide a more comprehensive assessment of reconstruction performance.

## 2 RELATED WORK

**Cryo-EM heterogeneous reconstruction.** Current methods for cryo-EM heterogeneous reconstruction can be broadly divided into non-neural and neural network-based approaches . 3D Classification (Scheres et al., 2007; Scheres, 2012; 2016; Punjani et al., 2017; Grant et al., 2018) employs the Expectation-Maximization algorithm to sort images into a predefined number of discrete classes (typically $< 10$) and is highly sensitive to initialization. Non-neural methods for continuous heterogeneity typically utilize linear models to address heterogeneity (Tagare et al., 2015; Andén & Singer, 2018; Punjani & Fleet, 2021; Gilles & Singer, 2023). 3DVA (Punjani & Fleet, 2021) and RECOVAR (Gilles & Singer, 2023) are PCA-based methods that use probabilistic PCA and regularized covariance estimation, respectively. These methods learn a linear subspace describing structural heterogeneity, but are limited in expressivity for diverse compositional heterogeneity settings (Jeon et al., 2024).

Neural network-based methods typically operate entirely in Fourier space, leveraging the Fourier slice theorem (Bracewell, 1956) for greater computational efficiency. CryoDRGN (Zhong et al., 2020) and Opus-DSD (Luo et al., 2023) are variational autoencoder (Kingma, 2013) (VAE)-based approaches that use MLP and CNN encoders and INR decoders, respectively, while SFBP (Kimanius et al., 2022) is a VAE whose decoder is a linear combination of voxel arrays. 3DFlex (Punjani & Fleet, 2023), DRGN-AI (Levy et al., 2024a), and Hydra (Levy et al., 2024b) are encoder-free autodecoder methods where each object has a learnable latent code. These existing methods mainly focus on conformational heterogeneity with one or two different species and share almost all of their decoder weights among all reconstructed structures. In contrast, our method focuses on extreme compositional heterogeneity and solves this problem by reducing parameter sharing via conditioning the INR by a hypernetwork, which dynamically adjusts the weights in every layer of the INR using a more powerful and scalable ViT (Dosovitskiy, 2020) encoder.

**Hypernetworks and INRs.** A hypernetwork (Ha et al., 2016) is a neural network $g_\phi$ that produces or modifies the weights of another neural network $f_\theta$, sometimes called the *primary network* or *hyponetwork*, typically an MLP, with the goal of learning the hypernetwork weights $\phi$. This architecture allows the weights of the primary network to be dynamically adapted to different tasks. Most forms of INR conditioning are equivalent to having a hypernetwork producing a subset of its weights (Xie et al., 2022). An important example is concatenation, the conditioning approach of most neural cryo-EM reconstruction methods, which is equivalent to defining an affine function that maps latent codes to the biases of the first layer of the network (Sitzmann et al., 2020a; Dumoulin

et al., 2018; Mehta et al., 2021), making concatenation a special case of a hypernetwork with low expressivity and high parameter sharing. In between the expressivity of full hypernetworks and concatenation are methods that predict feature-wise transformations (Dumoulin et al., 2018; Chan et al., 2021; Mehta et al., 2021), also called FiLM (Perez et al., 2018) conditioning, which predict a per-layer scale and bias. Our method uses a hypernewtork architecture that predicts the weights (but not the biases) of each layer, making it more expressive than conditioning by concatenation or FiLM. Hypernetworks that produce the weights of the primary network directly are difficult to train (Ortiz et al., 2023), so often the weights of the primary network are modified using a residual learning approach (Chen & Wang, 2022; Ortiz et al., 2023). Hypernetworks have been widely used to condition INRs (Sitzmann et al., 2019; 2020b; 2021; Chen & Wang, 2022; Gu et al., 2023; Kim et al., 2023; Lee et al., 2024; Gu & Yeung-Levy, 2025), especially generalizable INRs (Chen & Wang, 2022; Kim et al., 2023; Gu et al., 2023; Lee et al., 2024; Gu & Yeung-Levy, 2025), where they outperform alternative methods of conditioning INRs such as gradient-based meta-learning (Tancik et al., 2021). In particular, Chen & Wang (2022) proposed a transformer-based hypernetwork architecture that uses a ViT encoder to modify the weights of an INR decoder via masking. Our insight is that these methods are designed to handle extreme compositional heterogeneity in shapes, with our method building on Chen & Wang (2022) by adapting it to the task of cryo-EM reconstruction.

**Heterogeneous benchmarks for cryo-EM.** The main heterogeneous benchmark for cryo-EM reconstruction is CryoBench (Jeon et al., 2024), which proposes five new datasets with varying types of heterogeneity and degrees of difficulty. Among these, Tomotwin-100 is the only CryoBench dataset that tackles extreme compositional heterogeneity with 100 distinct structures. We extend this further by proposing Sim2Struct-1000, a large-scale and challenging dataset for compositional heterogeneity derived from Giri et al. (2024) that has 10 times as many structures as Tomotwin-100. Additionally, the standard volume-based metric for cryo-EM is Fourier shell correlation (FSC), which can be misleading for heterogeneous structures (Gilles & Singer, 2023). To provide a more complete evaluation of reconstruction quality, we propose two complementary real-space metrics that can capture heterogeneity missed by FSC.

## 3 METHODS

In this section, we introduce the cryo-EM image formation model (Section 3.1), motivation (Section 3.2), and our transformer-based hypernetwork method, CryoHype (Section 3.3).

### 3.1 CRYO-EM IMAGE FORMATION MODEL

The cryo-EM reconstruction task is to recover structures $V_i : \mathbb{R}^3 \to \mathbb{R}, 1 \le i \le N$ of a set of noisy 2D projections $X_1, \ldots, X_N$ of the structures $V_i$. In each projection $X_i$, the particle is in an unknown pose $\phi_i$, consisting of a rotation $R \in SO(3)$ and in-plane translation $t \in \mathbb{R}^2$. Each image $X_i$ is generated according to the following model:

$$X_i = C_i * \mathcal{P}(\phi_i)V_i + \epsilon \tag{1}$$

where $C_i$ is the Contrast Transfer Function (CTF), $\mathcal{P}$ is the projection operator that transforms $V_i$ by rotation by $R_i$ and translation by $t_i$, and $\epsilon \sim \mathcal{N}(0, \sigma^2)$ models additive isotropic Gaussian noise. Additional details are provided in the Appendix.

### 3.2 MOTIVATION

Previous neural volume representation methods in cryo-EM captured heterogeneity through either providing a latent code as additional input to an implicit neural representation (INR) volume representation (Zhong et al., 2020) or as as the coefficients of a linear combination of a shared basis of voxel arrays (Kimanius et al., 2022). In either of these approaches, almost all parameters of the neural volume representations are shared among all the different structures, limiting the diversity of the structures that can be captured and limiting the ability of the model to generate structure-specific high-resolution details. Hypernetworks overcome this problem by increasing the expressiveness of conditioning and reducing parameter sharing, since it can be proven that conditioning a network $\Psi$ by concatenation is equivalent to having a linear hypernetwork produce the biases of the first layer of $\Psi$ (Sitzmann et al., 2020a; Dumoulin et al., 2018; Mehta et al., 2021). Two observations stem

naturally from this result: first, that a general hypernetwork generalizes conditioning by concatenation and can be much more expressive if the hypernetwork is more expressive than a linear layer. Second, conditioning by concatenation is equivalent to sharing all hyponetwork (i.e. INR decoder) weights among all data points except its biases. Thus, hypernetwork approaches can dynamically adapt a significantly higher proportion of decoder weights than either conditioning-by-concatenation or linear combinations of voxel arrays.

### 3.3 CRYOHYPE ARCHITECTURE

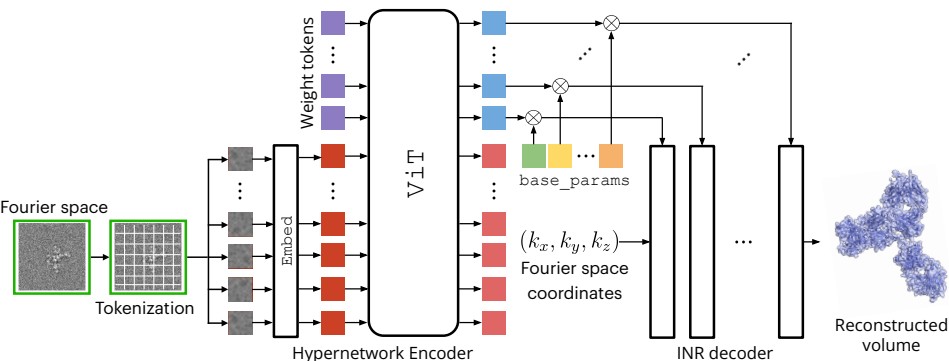

Figure 1: **CryoHype architecture**. First, the input image is tokenized using a ViT tokenizer `Embed`. They are concatenated with learnable weight tokens and processed by a `ViT` encoder. Using only the output (blue) tokens corresponding to the weight tokens, we apply linear heads $\text{Head}_i$ (not shown for clarity) and then dynamically adjust the base parameters of an INR decoder, which reconstructs the volume in Fourier space, using elementwise matrix multiplication $\otimes$. Finally, an inverse Fourier transform is applied to get the reconstruction in real space (not shown).

The CryoHype architecture consists of five main components: (1) a ViT encoder $g$, consisting of a tokenizer `Embed` and Transformer encoder `Enc`, (2) extra learnable weight tokens $\{w_i\}_{i=1}^q$ (3) an INR $f$, a ReLU MLP with residual connections, with a shared set of base parameters $\{\theta^i\}_{j=1}^L$ where $L$ is the number of layers, and (4) learnable linear heads $\{\text{Head}_j\}_{j=1}^L$ for each layer $L_j$ in $f$ (see Figure 1). Reconstruction is done completely in the Fourier domain. A forward pass of our model works as follows: first, an input projection $\hat{X}$ tokenized into $T$ tokens $\{t_k\}_{k=1}^T$ by `Embed`. These $T$ tokens are then concatenated along with the learnable weight tokens $w_i$ and processed by `Enc`, the Transformer part of the ViT encoder, to produce the final tokens $[t_1^F, \ldots, t_T^F, w_1^F, \ldots, w_q^F]$. The output tokens corresponding to the weight tokens $w_i^F$ are then divided into $L$ groups consisting of $a_j$ tokens $w_q^{F,j}, \ldots, w_{a_j}^{F,j}, 1 \le j \le L$, with $\sum_i a_j = q$. The $j$th group $[w_{a_1}^{F,j}, \ldots, w_{a_j}^{F,j}]$ is transformed by the linear head $\text{Head}_j$ and normalized. The output of the previous step is multiplied elementwise by the $j$th layer's base parameter $\theta_j$ to produce the final parameters $\theta_j^F$ of the $j$th layer::

$$\theta_i^F = \text{Norm}(\text{Head}_j([w_{a_1}^{F,i}, \ldots, w_{a_j}^{F,i}])) \otimes \theta_j \tag{2}$$

Finally, the final INR parameters $\theta_i^F$ are used to instantiate the INR $f$, which parametrizes the structure $\hat{V}$. The INR $f$ maps Fourier space coordinates $(k_x, k_y, k_z)$ to the Fourier-transformed electron scattering potential at that coordinate, producing a clean (i.e., not noisy and CTF-free) prediction $\tilde{X}$. $\tilde{X}$ is then multiplied by the CTF (see Section 3.1), and a reconstruction loss (mean-square error, MSE) is computed between the ground truth views and predicted views, and gradients are backpropagated to the hypernetwork. Note that CryoHype is trained end-to-end, with the learnable parameters being (1) the ViT encoder $g$, (2) the extra learnable weight tokens $w_i, 1 \le i \le q$, (3) the decoder's base parameters $\theta_j, 1 \le j \le L$, where $L$ is the number of layers in the decoder, and (4) the learnable linear heads $\text{Head}_j, 1 \le j \le L$.

**Latent space embeddings.** Unlike autoencoder and autodecoder-based reconstruction methods, CryoHype does not have a canonical low-dimensional latent space. For our latent space analysis, we use the tokens $w_1^F, \ldots, w_q^F$ (blue tokens of Fig. 1). These tokens have total dimension $qd$ where $q$ is

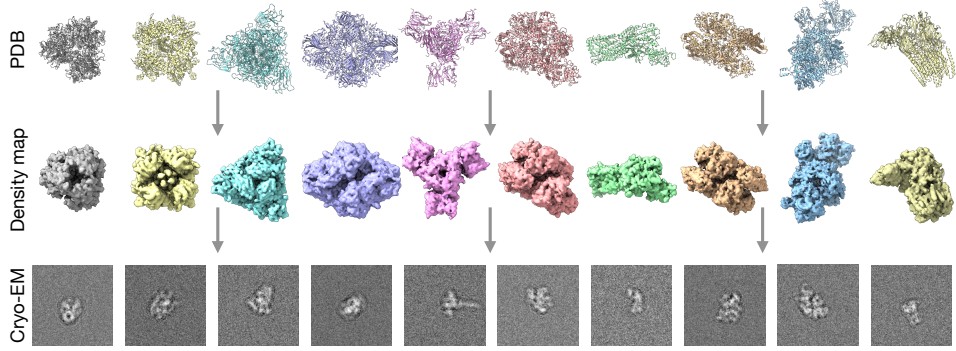

Figure 2: **Sim2Struct-1000.** Example atomic models, density maps, and projected images from `Sim2Struct-1000`, containing 1000 distinct structures.

the number of weight tokens and $d$ is the dimension of the ViT and are extremely high-dimensional. To get an interpretable latent space, we perform dimensionality reduction in two stages: first, we use principal component analysis (PCA) to reduce to a smaller dimension $d_1 \ll qd$, with $d_1 = 100$. We then use UMAP (McInnes et al., 2018) to further reduce the dimension to 2 for visualization.

## 4 SIM2STRUCT-1000

We introduce Sim2Struct-1000, a large-scale simulated cryo-EM dataset for extreme compositional heterogeneity derived from the Cryo2StructData collection (Giri et al., 2024). Cryo2StructData comprises experimentally obtained cryo-EM density maps paired with atomic models from the Protein Data Bank (PDB) (Berman et al., 2000). Experimental cryo-EM maps from the original collection exhibited inconsistent resolution, noise levels, and grid dimensions due to diverse experimental parameters, potentially introducing confounding downstream biases. To avoid training models that learn these experimental settings, we instead selected a subset of 1000 atomic models filtered by particle size for Sim2Struct-1000. Each atomic model was converted to a density map and subsequently projected to create 1000 simulated images ($256 \times 256$, 3.0 Å/pix, downsampled to $128 \times 128$), resulting in a dataset of 1M total particle images (Figure 2). Sim2Struct-1000 thus allows evaluation of our method's robustness under challenging conditions of compositional heterogeneity at scale.

## 5 EXPERIMENTAL SETTINGS

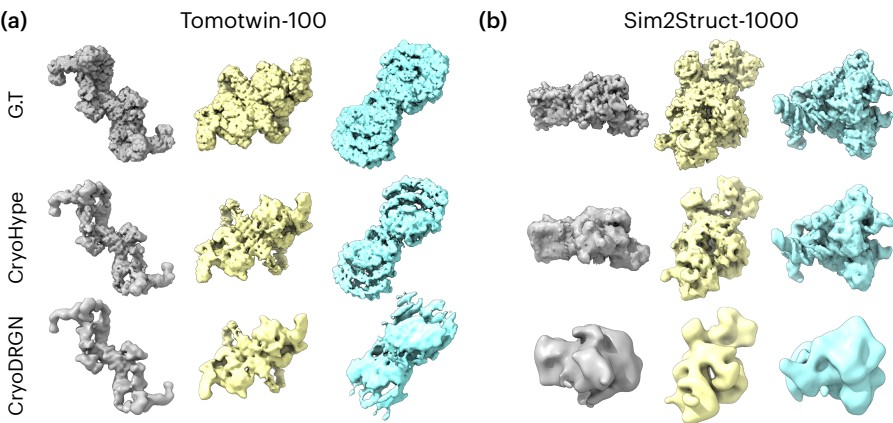

Figure 3: **Qualitative results of Tomotwin-100 and Sim2Struct-1000.** Representative density volumes and the corresponding ground truth volume.

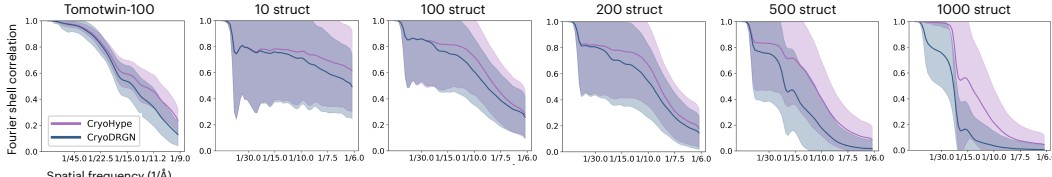

Figure 4: **Per-Image FSC.** Each curve shows the average FSC curve across all conformations with error bars indicating the standard deviation. The full FSC curves are shown in Appendix.

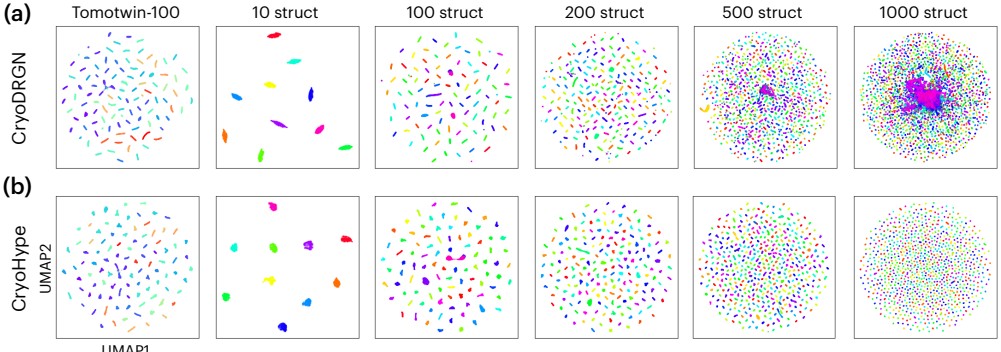

Figure 5: **Latent Visualization for Tomotwin-100 and Sim2Struct-1000.** (a) Latent embeddings from cryoDRGN visualized by UMAP and colored by the 10, 100, 200, 500, and 1000 G.T proteins. (b) Latent embeddings for CryoHype.

## 5.1 DATASETS

We evaluate our method on two heterogeneous synthetic datasets containing extreme compositional heterogeneity: Tomotwin-100 (Jeon et al., 2024) and our new challenging Sim2Struct-1000 dataset. We further demonstrate our method on one experimental dataset of the assembling ribosome (Davis et al., 2016). More details are in the Appendix.

**Tomotwin-100.** Tomotwin-100 (Jeon et al., 2024) evaluates the capability of cryo-EM reconstruction algorithms to address extreme compositional heterogeneity. This dataset was generated by simulating the cryo-EM image formation process for 100 of the 120 distinct cellular complexes included in the TomoTwin dataset (Rice et al., 2023), curated to contain diverse and dissimilar proteins. Notably, Tomotwin-100 represents the most challenging dataset in (Jeon et al., 2024), with most methods failing to achieve successful reconstructions. We also find that higher FSCs do not necessarily result in higher Chamfer Distance or volumetric IoU, indicating that our new metrics are capturing differences in structure that are not being captured by FSC.

**Sim2Struct-1000.** Sim2Struct-1000 evaluates model scalability to datasets containing a large degree of compositional heterogeneity. Sim2Struct-1000 is a synthetic dataset dervied from Cryo2Struct (Giri et al., 2024) (see Sec. 4). In our experiments, we examine four subsets of this dataset, representing different amounts of compositional heterogeneity, consisting of 10, 100, 200, 500, and all 1000 structures. Each structure has 1000 simulated projection images.

**EMPIAR-10076.** We also evaluate our method on an experimental dataset, EMPIAR-10076 (Davis et al., 2016), which is known to exhibit significant compositional

Table 1: **Quantitative performance on Tomotwin-100 (Noiseless)**, measured by $\text{FSC}_{\text{AUC}}$, CD, and vIoU. Metrics computed on backprojected images for each G.T. structure as an upper bound.

| Method | Noiseless Tomotwin-100 | | | | | |
|---|---|---|---|---|---|---|
| | ↑ Mean $\text{FSC}_{\text{AUC}}$ (std) | Median | ↓ Mean CD (std) | Med | ↑ Mean vIoU (std) | Med |
| CryoDRGN Zhong et al. (2021) | 0.328 (0.022) | 0.327 | 1.9750 (0.4450) | 1.9165 | 0.6513 (0.0540) | **0.6534** |
| **CryoHype** | **0.384 (0.019)** | **0.387** | **1.8663 (0.2514)** | **1.9002** | **0.6564 (0.0375)** | 0.6512 |
| Backprojection | 0.406 (0.018) | 0.406 | 1.1931 (0.1987) | 1.2130 | 0.7527 (0.0406) | 0.7500 |

Table 2: **Quantitative performance on Tomotwin-100 (Noisy), measured by** $\text{FSC}_{\text{AUC}}$**, CD, and vIoU.** † indicates that the result is from Jeon et al. (2024).

| Method | Tomotwin-100 | | | | | |
|---|---|---|---|---|---|---|
| | ↑ Mean $\text{FSC}_{\text{AUC}}$ (std) | Median | ↓ Mean CD (std) | Med | ↑ Mean vIoU (std) | Med |
| CryoDRGN (Zhong et al., 2021) | 0.316 (0.046)† | 0.321† | 2.26 (1.59) | **1.98** | **0.63 (0.08)** | **0.65** |
| DRGN-AI-fixed (Levy et al., 2024a) | 0.202 (0.044)† | 0.207† | 32.60 (18.45) | 29.52 | 0.13 (0.09) | 0.12 |
| Opus-DSD (Luo et al., 2023) | 0.237 (0.049)† | 0.251† | 33.48 (0.1378) | 28.92 | 0.14 (0.08) | 0.13 |
| SFBP (Kimanius et al., 2022) | 0.036 (0.011) | 0.036 | 18.52 (8.33) | 17.32 | 0.16 (0.06) | 0.16 |
| 3DVA (Punjani & Fleet, 2021) | 0.088 (0.040)† | 0.077† | 25.52 (17.90) | 21.40 | 0.18 (0.09) | 0.18 |
| RECOVAR (Gilles & Singer, 2023) | 0.258 (0.109)† | 0.254† | 27.22 (18.86) | 23.14 | 0.16 (0.08) | 0.15 |
| 3D Class (Punjani et al., 2017) | 0.046 (0.026)† | 0.037† | - | - | - | - |
| **CryoHype** | **0.346 (0.033)** | **0.353** | **2.18 (0.46)** | 2.11 | 0.61 (0.06) | 0.62 |
| Backprojection | 0.364 (0.023) | 0.364 | 1.50 (0.20) | 1.50 | 0.71 (0.03) | 0.71 |

Table 3: **All Sim2Struct-1000 metrics.** Metrics are computed with standard deviations per method in parentheses. Chamfer distance is given in angstroms (Å). Isosurface levels are set at 220 for all subsets of `Sim2Struct-1000`.

| Method | Structures | Sim2Struct-1000 | | | | | |
|---|---|---|---|---|---|---|---|
| | | ↑ Mean $\text{FSC}_{\text{AUC}}$ (std) | Median | ↓ Mean CD (std) | Median | ↑ Mean vIoU (std) | Median |
| CryoDRGN | 10 | 0.434 (0.012) | 0.437 | 1.9898 (0.3010) | 2.0468 | 0.4853 (0.0524) | 0.4806 |
| CryoHype | | **0.464 (0.006)** | **0.465** | **1.7781 (0.1702)** | **1.7890** | **0.5005 (0.0336)** | **0.4939** |
| CryoDRGN | 100 | 0.361 (0.039) | 0.357 | 2.3389 (0.6433) | 2.2417 | 0.4731 (0.0602) | 0.4664 |
| CryoHype | | **0.409 (0.024)** | **0.407** | **1.9916 (0.4040)** | **1.9488** | **0.4897 (0.0516)** | **0.4849** |
| CryoDRGN | 200 | 0.334 (0.047) | 0.334 | 2.4428 (1.0553) | 2.2273 | **0.4765 (0.0673)** | **0.4766** |
| CryoHype | | **0.377 (0.028)** | **0.375** | **2.0748 (0.3363)** | **2.0489** | 0.4726 (0.0484) | 0.4697 |
| CryoDRGN | 500 | 0.216 (0.069) | 0.213 | 4.6358 (4.2948) | 3.1548 | 0.3866 (0.1293) | 0.4101 |
| CryoHype | | **0.305 (0.065)** | **0.322** | **2.4069 (0.7773)** | **2.2336** | **0.4529 (0.0773)** | **0.4565** |
| CryoDRGN | 1000 | 0.139 (0.054) | 0.140 | 9.0656 (7.6560) | 5.9439 | 0.2647 (0.1406) | 0.2608 |
| CryoHype | | **0.232 (0.079)** | **0.216** | **3.0179 (1.2470)** | **2.6512** | **0.4181 (0.1088)** | **0.4394** |

heterogeneity, comprising 13 discrete structures of the assembling 50S ribosome organized into four major assembly states. The data is preprocessed according to Zhong et al. (2021).

## 5.2 METRICS

We measure reconstruction quality with three metrics. The first, Fourier Shell Correlation (FSC), is a standard metric for comparing volumes in cryo-EM, computing correlation between Fourier shells at various thresholds and is a global measure of resolution, but can be misleading in the heterogeneous case (Gilles & Singer, 2023). We follow (Jeon et al., 2024) and evaluate methods using the area under the FSC curve per image ($\text{FSC}_{\text{AUC}}$). In addition, we propose two metrics from 3D shape analysis that measure reconstruction quality in real space and thus are more sensitive to local structural heterogeneity. The first is volumetric intersection-over-union (IoU), which measures the volumetric overlap between volumes, and the second is Chamfer distance (CD), which captures pointwise differences between point clouds. We convert the voxel-based data to point clouds by extracting the coordinates of occupied voxels above a specified density threshold and scaling these to world coordinates based on voxel size and grid dimensions. Further analyses of our new metrics and details on the density threshold selection can be found in the Appendix.

## 5.3 BASELINES

In Table 2, we examine a variety of state-of-the-art fixed-pose methods for cryo-EM reconstruction, including VAE-based reconstruction algorithms (Zhong et al., 2021; Kimanius et al., 2022; Luo et al., 2023), autodecoder-based algorithms (Levy et al., 2024a), and non-deep learning based algorithms (Punjani et al., 2017; Gilles & Singer, 2023; Punjani & Fleet, 2021). For other experiments, we only compare against CryoDRGN (Zhong et al., 2021), the only method that demonstrates reasonable performance in the case of extreme compositional heterogeneity (Jeon et al., 2024).

## 6 RESULTS

We evaluate our model on `Tomotwin-100` and our new `Sim2Struct-1000` dataset. For synthetic datasets with ground truth, quantitative results including $\text{FSC}_{\text{AUC}}$, Chamfer Distance, and

volumetric IoU are found in Tables 1, 2, and 3. Qualitative results for the Tomotwin-100 and Sim2Struct-1000 datasets are found in Figure 3.

**Tomotwin-100.** In the noiseless case, we find that CryoHype greatly outperforms CryoDRGN in $\text{FSC}_{\text{AUC}}$, approaching the performance of backprojection, and is better or comparable in all real space 3D shape metrics (Table 1). In the standard noisy (SNR 0.01) case, we note that all baselines except CryoDRGN are unable to handle extreme compositional heterogeneity and fail to produce reasonable reconstructions (Jeon et al., 2024) (Table 2). Here, we find that CryoHype clearly outperforms all baselines, including CryoDRGN, in the standard $\text{FSC}_{\text{AUC}}$ metric, but exhibits more mixed performance in 3D shape metrics when compared to CryoDRGN. Qualitatively, we find that CryoHype captures the global shape more precisely and more fine-grained details than CryoDRGN, resulting in higher resolution (Figure 3(a)), which is also reflected in the FSC curves (Figure 4). We attribute this to the more expressive conditioning of the CryoHype's hypernetwork architecture. Both CryoHype and CryoDRGN produce reasonable looking latent spaces (Figure 5). We also find that CryoHype has much less variability, as indicated by smaller standard deviations for all metrics.

**Sim2Struct-1000.** Quantitatively, we find that CryoHype significantly outperforms CryoDRGN at all levels of compositional heterogeneity (10, 100, 200, 500, and 1000 structures) in virtually all metrics, including our proposed 3D shape metrics (Table 3). As shown by the FSC curves, CryoHype captures all frequencies better than CryoDRGN across all levels of compositional heterogeneity (Figure 4). Qualitatively, we see the same behavior as Tomotwin-100, where CryoHype produces high resolution reconstructions due to its more expressive conditioning of the INR decoder, while CryoDRGN is oversmoothing (Figure 3). We also find that CryoHype's performance advantage over CryoDRGN increases as the compositional heterogeneity gets more extreme, showing the better scaling of our ViT encoder vs CryoDRGN's MLP encoder. This trend is also reflected in the latent spaces (Figure 5). We find that while the latent spaces for both methods look reasonable at lower levels of heterogeneity, the latent space of CryoDRGN starts to degrade at high levels of heterogeneity (500 structures and 1000 structure), indicating that CryoDRGN can no longer completely resolve the heterogeneity in the dataset. In contrast, the latent space of CryoHype remains clustered for views of the same structure and disentangled for different structures, even at the most extreme amounts of compositional heterogeneity.

**EMPIAR-10076.** Figure 6(a) illustrates reconstructed volumes of the four major classes of `EMPIAR-10076` produced by CryoHype and CryoDRGN. Due to the absence of ground truth volumes for this dataset, direct comparison of quality between the methods is challenging. Instead of quantitative metrics, we present latent space visualizations colored by major and minor classes identified from the original publication (Fig. 6 (b)). Both methods successfully separate the major classes. However, for the minor classes, CryoHype produces distinct clusters within each major class (e.g., D1, D2, D3, and D4), while CryoDRGN shows considerable overlap and less distinct separation among the clusters.

Table 4: Ablation study on CryoHype examining the four main components of the model, evaluated by $\text{FSC}_{\text{AUC}}$.

| Method | Tomotwin-100 | |
|---|---|---|
| | Mean (std) | Med |
| Concatenation | 0.255 (0.076) | 0.286 |
| U-Net encoder | 0.208 (0.031) | 0.214 |
| MLP encoder | 0.234 (0.032) | 0.240 |
| **CryoHype** | **0.346 (0.033)** | **0.353** |

## 6.1 ABLATION

In Table 5, we show the effectiveness of the hypernetwork architecture and ViT encoder of CryoHype. We find that changing to a different encoder results in heavily degraded performance, confirming that the hypernetwork architecture is more expressive. To test the effectiveness of the ViT encoder, we replace it with a convolutional U-Net (Ronneberger et al., 2015; Buda et al., 2019) encoder and MLP encoder (Sitzmann et al., 2021). Performance is again heavily degraded despite the convolutional and MLP networks using more parameters, showing the importance of using a ViT encoder in hypernetwork architectures for parameter efficiency and scalability. Additional experiments comparing CryoHype against larger CryoDRGN variants as well full implementation details can be found in the Appendix.

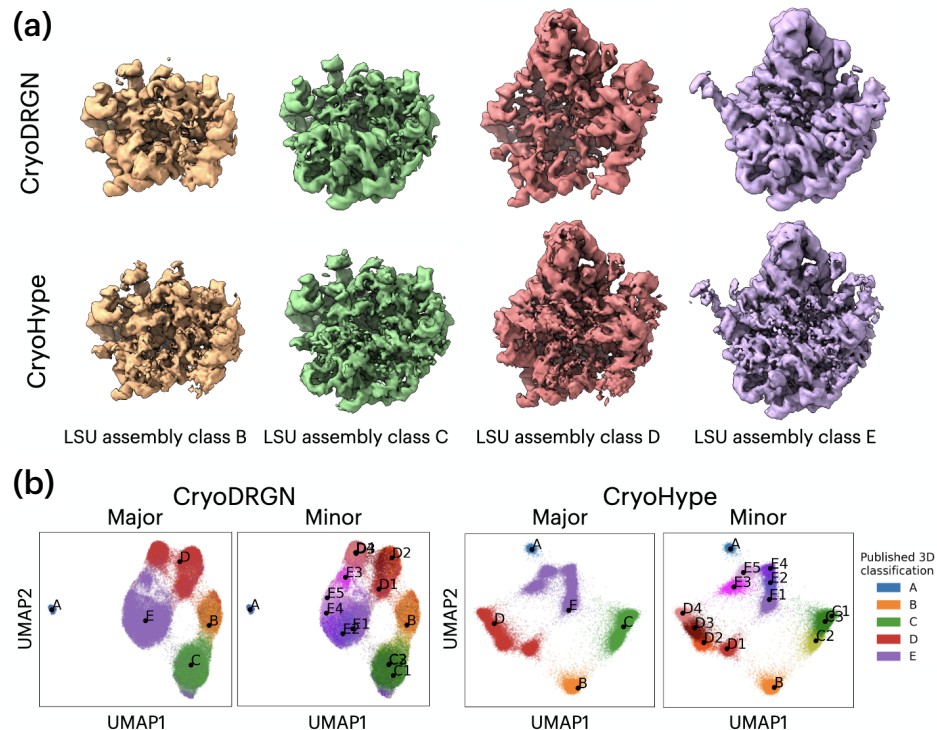

Figure 6: **Qualitative results on the EMPIAR-10076 dataset.** (a) Density maps of the four major ribosome assembly states from Davis et al. (2016). (b) Latent space representation, colored by major and minor assembly states assigned from the 3D classification in Davis et al. (2016). CryoHype produces a disentangled latent space in which major and minor classes are well-clustered, facilitating clear separation and classification.

## 7 CONCLUSION

We introduce CryoHype, a novel transformer hypernetwork approach for cryo-EM reconstruction that can dynamically adapt the decoder to each input image, allowing our method to capture large-scale compositional heterogeneity at high resolution. Across both synthetic and experimental datasets, we show that CryoHype can more accurately recover compositional heterogeneity from large-scale datasets over previous methods and produces more structured latent spaces. We also introduce `Sim2Struct-1000`, a new dataset for compositional heterogeneity with 10 times more structures than existing datasets, as well as two complementary real-space metrics for evaluating cryo-EM reconstruction quality.

In this work, we focus on the architectural expressivity of hypernetworks for modeling extreme-scale compositional heterogeneity, and we note that CryoHype currently requires known particle poses. While this assumption is unrealistic in real experimental settings, it allows us to isolate and study the benefits of transformer-based hypernetwork conditioning. Extending CryoHype to *ab initio* reconstruction with joint pose estimation is an important next step, with natural integration into existing pose-search frameworks. Beyond poses, future work could investigate datasets containing both conformational and compositional heterogeneity, motion recovery within the latent space, and multi-view extensions such as tilt-series imaging. Our two new metrics (CD, IoU) are sensitive to isosurface levels, indicating a need for future metrics that are independent of isosurface or noise factors. Together, these advances suggest that transformer-based hypernetworks, coupled with large-scale heterogeneous datasets, offer a foundation for developing computational methods to enable reconstructing diverse mixtures from cryo-EM at scale.

**Ethics statement.** Our method for reconstructing the conformations of biomolecules from cryo-EM imaging should help increase the understanding of the biological functions of the reconstructed

biomolecules. We do not believe that our method has any negative societal impacts. LLM usage was limited to improving the writing of the paper.

**Reproducibility statement.** We will release the code and our new `Sim2Struct-1000` dataset upon publication.

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
