# OpenReview forum: "CryoHype: Transformer-based hypernetworks for extreme heterogeneity in cryo-EM"
_ICLR.cc/2026/Conference — ICLR 2026 Conference Withdrawn Submission_

### Official Review · Reviewer_Dog6 · 2025-10-19

**Soundness:** 2
**Presentation:** 2
**Contribution:** 3
**Rating:** 4
**Confidence:** 4

**Summary:**

The authors introduced CryoHype, a hypernetwork-based cryo-EM reconstruction method for complex compositional heterogeneity. The method utilizes a ViT encoder to encode the heterogeneity information and adopts a hypernetwork design to use the encoded information to condition the cryo-EM density. Experimental results show that CryoHype can accurately reconstruct the heterogeneity states, and on simulated datasets, its performance surpasses all baselines.

**Strengths:**

- The adoption of hypernet in cryo-EM reconstruction is relatively new.
- The reconstructed compositional heterogeneity states shown in the paper have high quality and are well separated from each other.
- The introduction of the Sim2Struct-1000 dataset is beneficial for benchmarking and developing future methods.

**Weaknesses:**

- The notation in the description of the method (Section 3.2) is confusing.
- The necessity of hypernetwork design is not justified. The only ablation study in Table 4 is about various encoders and aims to show that ViT is the right choice. But there are many other ways to use the encoded information to condition the cryo-EM density, and these designs are not compared in the paper.
- The only evaluated experimental dataset is EMPIAR-10076, which is commonly used but may not be sufficient to demonstrate the effectiveness of the method on more experimental cases.
- While the dataset could be helpful for the community, the noise level shown in Figure 2 might be too mild.

**Questions:**

- Since there are $q$ weight tokens, what does $w_q^{F,j}, \ldots, w_{a_j}^{F,j}$ in Line 199 mean? My understanding from Figure 1 is that each weight token after passing the ViT will be used to condition one INR decoder layer.
- What would the performance be like if decoder designs other than hypernetworks are adopted? For example, one intuitive way is to add a transformer decoder that takes in the weight token for cross attention and also takes in the Fourier space coordinates as queries.
- How would the current pipeline’s performance be on the SARS-CoV-2 spike protein dataset and EMPIAR-11043 dataset?

---

### Official Review · Reviewer_S8bX · 2025-11-01

**Soundness:** 2
**Presentation:** 3
**Contribution:** 3
**Rating:** 2
**Confidence:** 5

**Summary:**

The paper introduced CryoHype, a method that uses transformer-based hypernetworks to resolve datasets with extreme (>100) compositional heterogeneity. The paper also presented a synthetic dataset under the extreme heterogeneity setting. CryoHype outperforms other methods including cryoDRGN on the synthetic benchmark datasets.

**Strengths:**

- The introduction of hypernetworks to cryo-EM reconstruction is novel and interesting.
- The newly simulated dataset in this paper is valuable and can be used as a benchmark dataset for similar tasks.

**Weaknesses:**

- My main critique for this paper is that, extreme heterogeneity (and the majority of the heterogeneity is compositional) with fairly accurate particle poses is not a real setting in cryo-EM. Samples with great compositional heterogeneity are indeed difficult to process in many steps in the cryo-EM processing pipeline, but the particle poses are unknown. However, the formulated problem in the paper is an extreme heterogeneity particle set (with even hundreds or a thousand classes) with known and fixed poses. In my opinion, although CryoHype shows better performance than other methods under this setting, such setting does not exist in real experiments.

- Minor: In Fig. 6(a) and Fig. 18, the reconstructed maps look much noisier than cryoDRGN's result.

**Questions:**

- In the two experiments with synthetic datasets, since the ground truth density maps, the classification for each particles are already known, is it possible to compute some metrics by comparing the results to the ground truth? E.g. the FSC between the reconstructed densities and GT, the accuracy of assigning particles to their correct classes/conformations?

---

### Official Review · Reviewer_xR5t · 2025-11-01

**Soundness:** 2
**Presentation:** 2
**Contribution:** 2
**Rating:** 4
**Confidence:** 5

**Summary:**

The paper proposes a transformer-based approach for heterogeneous reconstruction problem in cryo-EM. The proposed method, referred to as CryoHype, uses hypernetworks and implicit neural representations to reconstruct cryo-EM density volumes from aligned submicrographs. The paper also proposed a highly compositionally diverse dataset with 1000 distinct structures, where it showed clear superiority over baseline methods for the task.

**Strengths:**

+The paper is overall well written and comprehensible
+The paper has both method and dataset contribution
+Extensive experiments across different datasets were performed.

**Weaknesses:**

- For single-particle cryo-EM, a high compositional diversity simply does not exist in real world. So having good results on artificially simulated highly diverse structure single particle cryo-EM dataset does not make much sense in terms of real use-case.
- In the only real dataset (EMPIAR-10076), the performance of cryoDRGN and the proposed method is similar. The paper mentions that CryoHype produces more distinct clusters within each major class than cryoDRGN, but does not show it quantitatively.
- The method works when poses are known. Unknown poses is actually the fundamental challenge in cryo-EM reconstruction. There is no discussion how the method performs in ab-initio cases where poses are unknown.
-There are several minor presentation issues, such as, the explanation of the word INR comes at line 154, though it is used from the beginning.

**Questions:**

What is the noise level in the Sim2Struct dataset?
Can you provide examples of real-world cryo-EM use cases, where a highly compositionally diverse structures are present?
How does CryoHype perform for ab-initio reconstruction?

---

### Official Review · Reviewer_t2Kd · 2025-11-02

**Soundness:** 2
**Presentation:** 2
**Contribution:** 1
**Rating:** 0
**Confidence:** 5

**Summary:**

This paper proposed CryoHype, a transformer-based hypernetwork for cryo-EM reconstruction that dynamically adjusts the weights of an implicit neural representation conditioned on each particle image. It demonstrated impressive performances on different datasets.

**Strengths:**

This paper proposed a hyper network to achieve NeRF on cryo-EM datasets.

**Weaknesses:**

1. This paper follows the NeRF idea for cryo-EM reconstruction but without citing and comparison with CryoNeRF (https://www.biorxiv.org/content/10.1101/2025.01.10.632460v1), which is the early work on cryo-EM reconstruction. I think the authors failed to do enough literature research. Without the comparison, I can not know the contributions and improvements over CryoNeRF.

2. This work did not present the performances on the basic settings: the homogeneous setting of cryo-EM reconstruction

3. This work did not study the simpler case but also important scenario: conformational cryo-EM reconstruction.

**Questions:**

1. The comparisons with SOTA work cryoNeRF
2. Generalizability to other cryo-EM reconstruction settings.

---

### Note · Authors · 2025-11-14

I have read and agree with the venue's withdrawal policy on behalf of myself and my co-authors.